# Geometric transformation adaptive optics (GTAO) for volumetric deep brain imaging through gradient-index lenses

Yuting Li[1,2,4], Zongyue Cheng [1,2,4], Chenmao Wang [1,2], Jianian Lin [1,2], Hehai Jiang[1,2] & Meng Cui [1,2,3] ✉

The advance of genetic function indicators has enabled the observation of neuronal activities at single-cell resolutions. A major challenge for the applications on mammalian brains is the limited optical access depth. Currently, the method of choice to access deep brain structures is to insert miniature optical components. Among these validated miniature optics, the gradient-index (GRIN) lens has been widely employed for its compactness and simplicity. However, due to strong fourth-order astigmatism, GRIN lenses suffer from a small imaging field of view, which severely limits the measurement throughput and success rate. To overcome these challenges, we developed geometric transformation adaptive optics (GTAO), which enables adaptable achromatic large-volume correction through GRIN lenses. We demonstrate its major advances through in vivo structural and functional imaging of mouse brains. The results suggest that GTAO can serve as a versatile solution to enable large-volume recording of deep brain structures and activities through GRIN lenses.

To understand the principles of neural networks, it is of great importance to observe their activities at cellular and subcellular spatial resolutions in the brains of awake and behaving animals[1–4]. Thanks to the rapid advance of genetic functional indicators[5–7] and the broad adoption of multiphoton fluorescence microscopy, such cellular functional recordings have become a routine practice in neuroscience research. Despite the rapid progress and adoption, these light-based measurements suffer from the shallow tissue access depth due to the spatially and temporally random light scattering in the mammalian brain tissues[8]. Currently, the noninvasive observations are limited to about one-millimeter depth[9,10]. Even for the centimeter-scale mouse brain, such access depth can only reach the outer region of the hippocampus and the majority of the brain volume is inaccessible. To study these inner brain regions, the common practice is to insert miniature imaging optics such as small diameter gradient-index (GRIN) lenses[11–23], miniature prisms[24,25], or their combinations[26,27]. For their compact form (e.g. 0.5 mm in diameter, 7 mm long), GRIN lenses gained popularity for deep brain functional recording. Through the

control of dopant concentrations, the GRIN lenses feature a radially varying index of refraction, which is a hyperbolic secant function of radial position. As a result, an on-axis focal spot can be repeatedly focused and relayed through propagation inside GRIN lenses[28]. For imaging applications, GRIN lenses suffer from severe optical aberration, dominated by fourth-order astigmatism[29]. The aberration in vector form can be expressed as $W_{222}(\vec{H}\vec{\rho})^2$, in which $W_{222}$ is the fourth-order aberration coefficient (the amount of aberration), $\vec{H}$ is the normalized field position and $\vec{\rho}$ is the normalized pupil position. For its quadratic dependence on the field position, astigmatism shoots up significantly in the outer region of the image field, significantly reducing the focal intensity and hence the excitation efficiency. Moreover, astigmatism notoriously features three focal planes (tangential, medial, and sagittal), greatly elongating the axial profile of the imaging system (Supplementary Fig. 1), which increases the fluorescence image background and diminishes the image contrast. For neural recording applications, astigmatism severely limits the usable field of view (FOV) and measurement throughput. If the cells of interest were missed by

---

[1]School of Electrical and Computer Engineering, Purdue University, West Lafayette, IN 47907, USA. [2]Bindley Bioscience Center, Purdue University, West Lafayette, IN 47907, USA. [3]Department of Biology, Purdue University, West Lafayette, IN 47907, USA. [4]These authors contributed equally: Yuting Li, Zongyue Cheng. ✉e-mail: mengcui@purdue.edu

the small recording FOV, new animals and surgeries will be required. Overall, the small functional imaging FOV, low measurement throughput and experimental success rate of GRIN lenses have become a major challenge for the study and understanding of deep regions of mammalian brains.

To overcome these challenges, optical correction methods have been explored[30–40]. A straightforward method is to apply a wavefront correction at the pupil plane of the objective lens[41–44]. A limitation of the pupil plane correction is that the fourth-order aberration has a quadratic dependence on the field position. Therefore, different regions demand different correction wavefront profiles. A static wavefront correction is only valid for a small region. A variation of this approach is to divide the entire measurement field of view into many subregions[45], dynamically switch the correction profile before the start of the imaging of each subregion, and digitally stitch these subregional images into the final image. Such a dynamic approach also suffers from two major problems. One is the imaging speed. Common commercial fast multiphoton microscopes are based on galvo scanners. Subregional scanning inherently suffers from reduced data throughput, undesirable for many applications[46]. Take calcium imaging, the most important and widely used application in neuroscience as an example, one will need to record from all accessible neurons nearly simultaneously at a high rate, which inherently demands large area or volume throughput. The other is the severe motion artifacts. During the measurement of awake and behaving animals, there is often substantial tissue motion. With conventional raster scanning, pixels in the adjacent rows and columns are recorded in close-by time points. Therefore, the image distortion due to tissue motion appears spatially continuous. With subregional scanning, the pixels across the subregional boundaries are discontinuous in time, causing strong motion artifacts. Due to these two drawbacks, spatially discrete correction and imaging are far from an ideal solution. A better strategy is to employ lens design to correct the astigmatism[47]. For example, an aspheric lens corrector has been developed to increase the FOV through GRIN lenses for multiphoton microscopy. With the spatially continuous correction, there is no change to the raster scanning and no loss in measurement throughput. Therefore, the method is robust for imaging awake and behaving animals. However, the demonstrated correction is not volumetric (correction effective for certain working distances) and the length of the GRIN lens corrected is insufficient for whole-brain access. In addition, the corrector lacks adaptivity to the variation of GRIN lenses and laser wavelengths. Overall, none of these explored solutions can meet all major practical requirements (throughput, volumetric correction, and adaptability) of in vivo volumetric functional imaging through the commonly used slim GRIN lenses (e.g. 0.5 mm in diameter, 7 mm long). In this work, we will present a strategy using off-the-shelf optical components to achieve volumetric correction for large-volume functional imaging through GRIN lenses, which is highly adaptive to GRIN lenses and laser wavelengths.

Our idea originated from the consideration of the nature of astigmatism, which is essentially the field curvature difference between the tangential and sagittal rays (Supplementary Fig. 1)[29]. Let's consider a beam that experienced astigmatism after propagating through the GRIN lens. If we can rotate the aberrated beam profile by 90 degrees such that the tangential and sagittal rays are interchanged and send the resulting transformed aberrated beam through a second GRIN lens, the field curvature difference should cancel out and the imaging system should be free from astigmatism. In such a way, we can use the transformed astigmatism to correct the original astigmatism. As this correction method is based on a 90-degree rotation of the beam spatial profile, we name this method geometric transformation adaptive optics (GTAO). In this study, we implemented GTAO through two sets of galvo scanners for two-photon fluorescence imaging and evaluated the efficacy of the GTAO imaging system under various experimental conditions. We demonstrated the significant advances of the GTAO system through structural and functional imaging of mouse brains.

## Results

### GTAO-based correction through 500 microns 1.5 pitch GRIN lenses

We first evaluated the idea through ray-tracing simulations (Fig. 1a). Let's consider two identical GRIN lenses of 500 microns in diameter and 1.5 pitch in length. With the odd number of half pitches, the input beam at 9 o'clock on one facet will exit at 3 o'clock on the output facet, aberrated by astigmatism through the GRIN lens. By positioning the aberrated beam at 12 o'clock on the input facet of a second GRIN lens, we effectively achieve the 90-degree beam rotation (interchanging the tangential and sagittal rays). The beam will exit at 6 o'clock on the output facet with astigmatism canceled. The ray tracing via Zemax showed that aberration experienced by a 930 nm laser with an input numerical aperture (NA) of 0.4 through the first GRIN lens reached 4.6 waves from the peak to the valley. In comparison, the value dropped to 0.9 waves through the GTAO-based correction. Furthermore, we decomposed the wavefront into Zernike modes (Supplementary Fig. 2a). As expected, astigmatism was indeed the dominant aberration term, which alone could reduce the Strehl ratio to ~ 10%. In comparison, the Strehl ratio in the presence of spherical aberration or coma was well above 90% (Supplementary Fig. 2b).

To evaluate GTAO experimentally, we set up a two-photon laser scanning microscope (Fig. 1b). Specifically, we employed two sets of two-axis galvo scanners (Saturn, ScannerMAX). The first scanner sent the 930 nm laser beam through an objective lens (LCPLN20XIR, Olympus) which coupled the light into the first GRIN lens. Its output was collected by another objective lens (LCPLN20XIR, Olympus). The second scanner achieved the function of 90-degree rotation (Supplementary Fig. 3) and delivered the beam into the imaging objective lens (HC FLUOTAR L 25x/0,95 W VISIR, Lecia) which coupled the laser into the GRIN lens implanted in the mouse brain. The fluorescence emission was also collected by the imaging objective lens and delivered to the photomultiplier tube (PMT, H10770PA, Hamamatsu) for detection. To eliminate the potential aberration contribution from the scanners, we employed a spatial light modulator (SLM, X10468-07, Hamamatsu) to display a static correction profile, which was measured for the on-axis beam only by a phase modulation-based AO technique[48,49]. The overall system's power throughput was 25% and the system group delay dispersion was ~12,000 $fs^2$ at 930 nm. To evaluate the focal profiles, we held a GRIN lens under the imaging objective lens and positioned a water immersion observation objective lens (Olympus 60x NA 1.2) under the GRIN lens, and used a camera to capture the laser focus across a 280-µm FOV (Fig. 1c, Supplementary Fig. 1). For comparison, we could turn off GTAO by simply keeping the laser beam static on the axis of the first GRIN lens while only using the second scanner to control the focus position. Without GTAO, the laser spot was diffraction-limited in the center of the FOV but degraded in the outer FOV by astigmatism. The maximum intensity projection (MIP) along the axial direction showed the signature cross shape caused by astigmatism[29]. In comparison, with GTAO the laser focus was still near round shape and its peak intensity was much greater at the outer region of FOV. Next, we embedded the 1.5 pitch GRIN lens inside agar with fluorescence beads of 1 µm in diameter and carried out two-photon fluorescence imaging of the beads through the GRIN lens. Similar to the one-photon camera-based observation, the image without GTAO was bright in the middle of the FOV but degraded in the outer regions (Fig. 1d). With GTAO, the beads appeared round and consistent over the entire FOV (Fig. 1e). The MIP of the image volumes also highlighted the impact of astigmatism and the performance of GTAO (Fig. 1f, g). In the outer region of the FOV, the image intensity without GTAO needs to be amplified by a factor of 10 to be visible in the same intensity display scale. Furthermore, the axial focal profile

was heavily stretched by astigmatism due to the offset between the sagittal and tangential focal planes (Fig. 1h, Supplementary Fig. 1). In comparison, the axial focal profile appeared rather consistent across the entire FOV with GTAO (Fig. 1i). To quantify the performance, we

extracted the cross-sectional plot (Fig. 1j, k) and measured the peak intensity ratio (Fig. 1l) and the transverse and axial profile ratios (Fig. 1m–o), which showed about an order of magnitude difference in the outer regions of the 280-μm FOV.

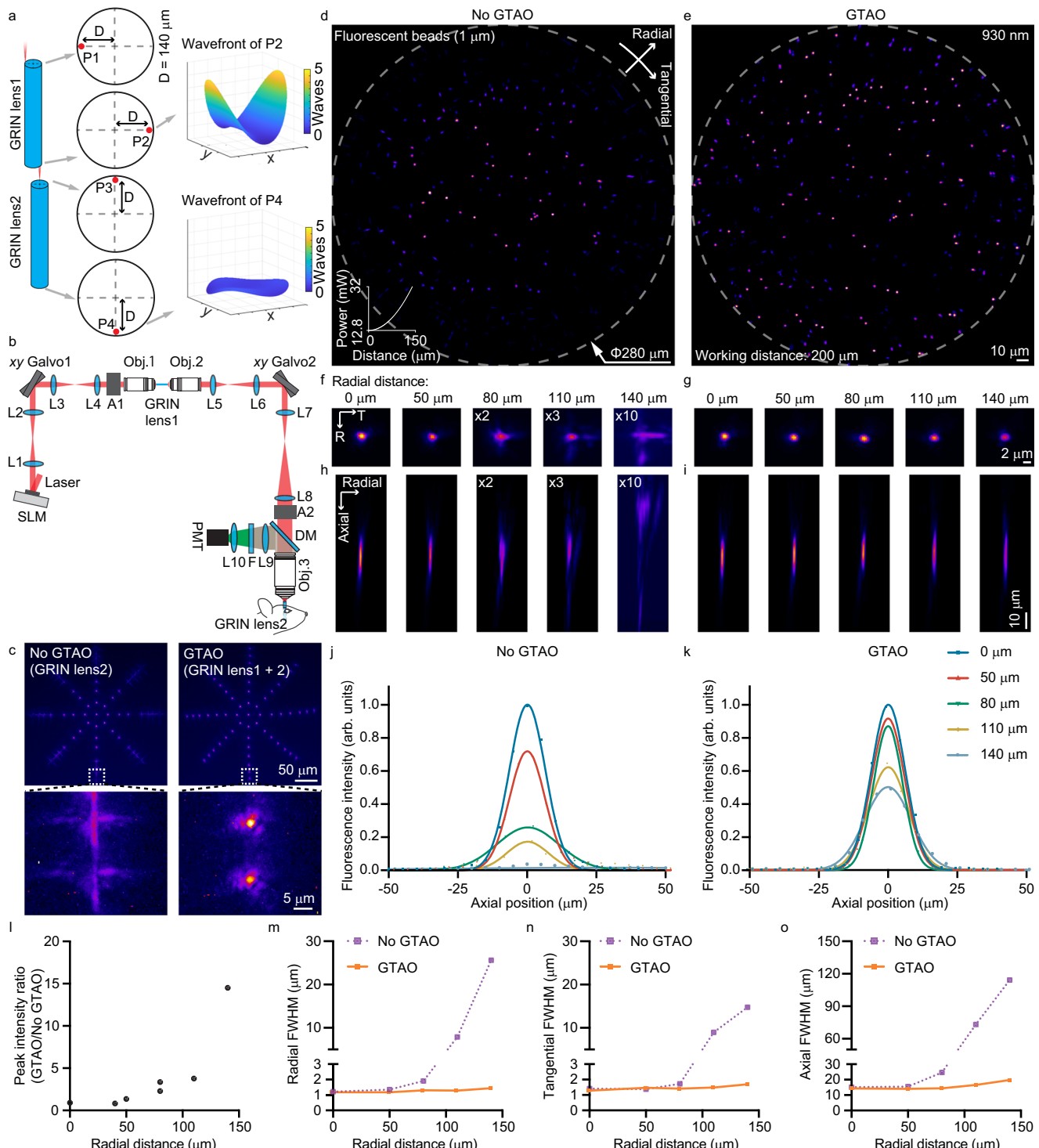

**Fig. 1 | System design of GTAO and point spread function quantification. a** The implementation of GTAO via a 90-degree spatial profile rotation and the associated optical wavefronts. **b** The optical design of the GTAO-based two-photon microscope. A1 and A2, magnetic mount which allowed the insertion of alignment optics; Obj, objective lens; L1-10, optical lenses; DM, dichroic mirror; PMT, photomultiplier tube; F, emission filter. **c** The focal profiles observed under GRIN lens 2 by a water immersion objective lens and a camera. The experiment was repeated

independently 5 times with similar results. **d, e** Two-photon fluorescence imaging of 1μm beads without and with GTAO, respectively. **f–i** The corresponding transverse and axial profiles of the bead images. The beads imaging was repeated independently 6 times with similar results. **j, k** The axial cross-sectional plot without and with GTAO, respectively. The beads imaging was repeated independently 3 times with similar results. **l–o** The comparison of peak intensity and spatial profiles between the images acquired with and without GTAO.

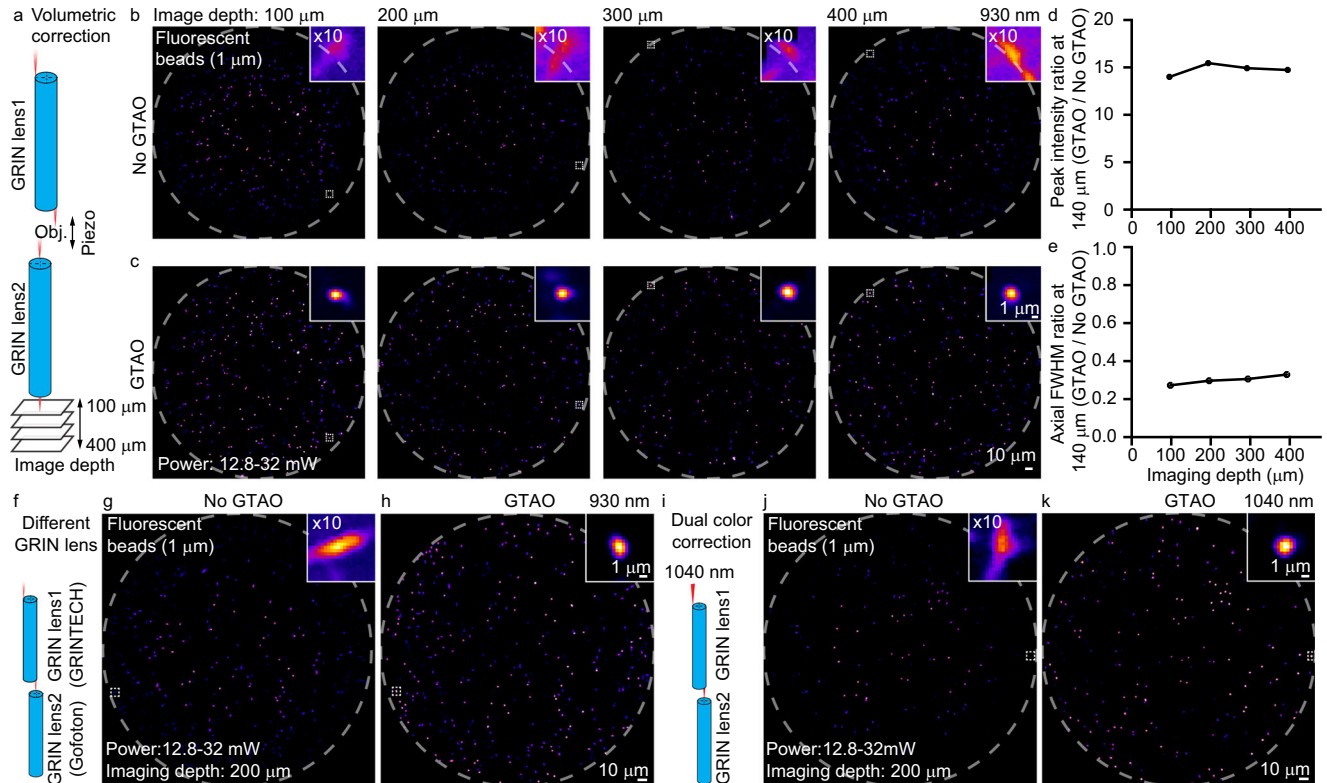

**Fig. 2 | The adaptability of GTAO to working distance, GRIN lens, and laser wavelength. a** The optical configuration for imaging at different working distances. **b, c** Two-photon images of 1 μm beads without and with GTAO at different working distances. The zoomed-in views of the beads in the dashed box were shown in the upper right corner. The beads imaging was repeated independently 4 times with similar results. **d, e** The ratio of peak intensity and axial FWHM in the outer regions of the FOV, respectively. **f** The optical configuration for using

GRINTECH lens to correct Gofoton lens. **g, h** Two-photon images of 1 μm beads without and with GTAO, respectively. The beads imaging was repeated independently 2 times with similar results. **i** The optical configuration for using the correction pattern established for 930 nm at 1040 nm. **j, k** Two-photon imaging of 1 μm beads at 1040 nm without and with GTAO, respectively. The beads imaging was repeated independently 2 times with similar results.

A key advance of the GTAO is the volumetric correction capability through GRIN lenses. In the common implementation of two-photon imaging, the users need to control the spacing between the imaging objective lens and the GRIN lens embedded in the mouse brain, which drives the spacing between the laser focus in the brain tissue and the GRIN lens facet. Over the typical working distance range (e.g. up to 400 μm for the 500 μm 1.5 pitch lens), the amount of astigmatism experienced by the laser beam is rather consistent (Supplementary Fig. 4a). Therefore, we can maintain the raster scanning pattern for both of the scanners while driving the imaging objective lens with a piezo scanner to achieve volumetric imaging and correction. To validate this idea, we employed the two-photon system to image 1 μm beads in agar at 100, 200, 300, and 400 μm working distance by only adjusting the imaging objective lens position (Fig. 2a). As expected, the GTAO consistently achieved high-quality correction. The bead images without GTAO need to be digitally amplified by a factor of 10 to be visible in the same intensity scale as the images acquired with GTAO (Fig. 2b, c). The peak intensity and axial FWHM also showed a rather consistent improvement (Fig. 2d, e). It was worth noting that GTAO improved the peak intensity by ~15 times in the outer regions of the FOV over the entire range of working distances.

In addition to working distances, GTAO is also adaptable to the variation of GRIN lenses. Lenses from different manufacturing batches or vendors which may have slight variations in NA and pitch length can be paired to achieve efficient correction. We developed an automated calibration procedure for pairing different lenses. Specifically, we used scanner 1 to position the laser spot on the outer edge of the FOV, used scanner 2 to undo this position shift, and then used

scanner 2 to move the laser spot in the orthogonal direction (90-degree rotation). The displacement from the FOV center controls the amount of astigmatism. We adjusted the displacement such that the laser spot at the exit facet of the second GRIN lens appeared round and symmetric and the peak focal intensity was maximized. The ratio of the laser spot displacement induced by scanners 1 and 2 was then recorded, which was subsequently used to generate the scanning control signal. Using ray tracing, we explored using two GRIN lenses of different NA for correction and confirmed high-quality corrections (Supplementary Fig. 4b–e). To experimentally demonstrate such adaptability, we used 500 μm diameter 1.5 pitch GRIN lenses from two lens makers (GRINTECH and GoFoton) to form the correction pair and performed two-photon imaging of 1 μm fluorescence beads. The data showed that pairing GRIN lenses from different manufacturers led to similar correction quality as that of the same manufacturers (Fig. 2f–h).

Besides adaptability for different lenses, GTAO also offers inherent achromatic astigmatism correction. The idea of GTAO is to have two orthogonally oriented astigmatism profiles to cancel the field curvature difference between tangential and sagittal rays, which is inherently not sensitive to wavelength (Supplementary Fig. 4f). Experimentally, we employed the same lenses and scanning parameters to image 1 μm fluorescence beads in agar at two excitation wavelengths. Specifically, the AO calibration was carried out for the 930 nm laser but the imaging was performed at 930 nm and at 1040 nm. At both wavelengths, GTAO achieved high-quality correction over the entire FOV (Fig. 2i–k). This experiment demonstrated the multi-color excitation capability of GTAO.

## GTAO-based two-photon structural imaging

To validate GTAO for neuroscience applications, we employed the two-photon system to image the brain slice of *Thy1*-eGFP mouse over a $300 \times 300\ \mu m^2$ FOV (Fig. 3a–f, Supplementary Movie 1). The diagonal dimension of the image reached 424 μm. Without GTAO, the cells appeared blurry beyond ~70 μm from the center of the FOV and became unresolvable at ~140 μm. In drastic comparison, with the assistance of GTAO, the cells were sharp over the 300 μm diameter and were still resolvable even in the corner regions. The cross-sectional plot (Fig. 3e) further highlighted the significant difference in image intensity and contrast (Fig. 3f) enabled by GTAO. Next, we employed the system to perform in vivo imaging of the *Thy1*-eGFP mouse brain (Fig. 3g–l, Supplementary Movie 2). In the outer region of the 334 μm imaging FOV, GTAO could resolve not only the somata but also the dendrites (Fig. 3j, k). With GTAO, the image contrast of soma was increased by ~10 times in the outer regions of the FOV (Fig. 3l). The difference for imaging dendrites was even greater as the strong astigmatism severely diminished the image contrast for fine structures.

Through in vivo structural imaging, we also evaluated the effectiveness of the on-axis correction provided by the SLM (Supplementary Fig. 5). Experimentally, we enabled and disabled SLM correction for GTAO and non-GTAO imaging. With the SLM correcting the on-axis aberration, the brightness and sharpness were improved for both cases. However, without GTAO, the cells in the outer regions (e.g. the ones encircled by the dashed line in Supplementary Fig. 5) remained blurry. In comparison, GTAO without SLM could still resolve these cells in the outer FOV.

An emerging application in neuroscience is to perform molecular imaging of the brain tissue to determine the cell types and identities after the in vivo functional recording[11]. For such studies, spatial registration between the image recorded through the GRIN lenses and the common objective lens becomes a critical step. Without GTAO, the cells in the outer regions of the FOV were blurry, making cellular registration challenging. With GTAO, the cells appeared in high contrast over the entire FOV Therefore, matching cells between the GRIN lens image and the objective lens image became much easier. However, there was a field curvature through GRIN lenses (Fig. 4a, b). Thus, after acquiring the image stack through the GRIN lenses, we performed a digital pixel shifting in the axial direction (Fig. 4c, d). After such digital field curvature correction, we could accurately and reliably match the images recorded by the GRIN lens and the objective lens (Fig. 4d, e). It is worth noting that we could achieve a consistent imaging FOV (Supplementary Fig. 6) with the same galvo driving signals through GRIN lenses of an integer number of half pitches (e.g. 1.5 pitch). For non-integer number of half pitches (e.g. 1.4 or 1.9 pitch), the FOV would increase at greater working distances with the same galvo driving signals (Supplementary Fig. 7). Nevertheless, the user could tailor the galvo driving signal range accordingly to achieve a consistent FOV when necessary.

## GTAO-based two-photon calcium imaging

As the most important and widely used application in neuroscience, calcium imaging demands near simultaneous large FOV volumetric high-throughput recording with adaptability to GRIN lenses and laser wavelengths. GTAO is specifically designed for such applications. We employed the GTAO two-photon system to perform in vivo calcium imaging of the mouse brain. To make a near real-time comparison of the imaging performance with and without GTAO, we designed the scanning control signals such that the odd number of image frames were recorded with scanner 1 parked in the center of the first GRIN lens and scanner 2 performing 2D raster scanning (no GTAO) and the even number of image frames were recorded with GTAO. The image frame rate was 10 Hz (5 Hz for each method). Through such a design, we can directly compare the signal level for the same calcium transients over

the 326-micron FOV (Fig. 5a). As expected, the in vivo imaging comparison showed much greater signal strength and image contrast with GTAO from both the somata and the dendrites (Fig. 5b–f, Supplementary Movie 3).

Next, we employed the GTAO system for volumetric calcium recording. As discussed previously, even without dynamically changing the scanner control signal, GTAO can enable volumetric correction (Fig. 2a–e). Experimentally, we drove the scanner at 2 kHz, which provided a 4 kHz line rate. With 200 lines per image frame, the frame rate was 20 Hz. Using a piezo stage to move the objective lens up and down with a 50 ms flyback time, we obtained 9 image planes over a total axial range of 160 μm at a 2 Hz volume rate (Supplementary Movie 4). As expected, GTAO enabled high-quality measurement for both somata and dendrites over the entire imaging volume (Fig. 5g–i).

## Discussion

One perspective on GTAO is that scanning through the first GRIN lens precisely acquires the wavefront which compensates the off-axis aberration generated by the second GRIN lens. Therefore, the aberration correction is spatially varying and temporally dynamic. In this manner, the field position-dependent astigmatism is dynamically and spatially continuously compensated. Such capabilities are beyond the reach of conventional pupil-plane AO, in which each wavefront is only valid for a small FOV. Thus, a large number of wavefront switching and image stitching would be required to synthesize the over 300-micron FOV provided by GTAO. Such segmented sub-FOV correction and recording would suffer from reduced measurement throughput and strong motion artifacts, which are unacceptable for practical neuroscience applications. Experimentally we compared the imaging performance with and without GTAO. It is worth noting that an SLM was employed to compensate for the static on-axis system aberration, which is the common implementation of pupil plane AO. Therefore, the comparison is essentially between the static pupil plane AO and the dynamic GTAO. As expected, the comparison study showed that the static correction is ineffective in addressing the strong astigmatism in the outer regions of the FOV while GTAO could more than double the high-quality FOV diameter and quadruple the area, which can significantly increase the functional imaging throughput and success rate in neuroscience applications. Compared to lens design-based correction, GTAO offers excellent adaptability. Firstly, GTAO can enable volumetric correction even without updating the scanning control. In fact, GTAO's scanning control signal can also be perfectly tailored for imaging at each working distance. But even without such variation, the correction was already of high quality. Secondly, GTAO can conveniently adapt to the variation of GRIN lenses. For example, the manufacturing process can have variations in pitch length from batch to batch. The GRIN lenses from different vendors may have different NA specifications. With GTAO, we can easily accommodate these changes by adjusting the scanner control signals (scaling the scanning range on the two GRIN lenses). The fourth-order astigmatism has a quadratic dependence on the field position. Therefore, we can fine-tune the amount of astigmatism by adjusting the scanning range on the first GRIN lens. For large variations such as changing the lens pitch number (e.g. 1, 1.5, 2), we can simply use a GRIN lens of the same pitch number to form the correction lens pair. Thirdly, GTAO's correction is not sensitive to wavelength. Using the same scanning control, we found that both the 930 nm and the 1040 nm lasers achieved optimal correction. A limitation of GTAO is that the correction is not instantaneously available to the entire FOV, which meets the requirements of laser scanning microscopy but not one-photon wide-field imaging. In our current implementation, the scanners are limited to non-resonant galvo scanners. The achieved maximum line rate was 4 kHz and the maximum frame rate was 20 Hz during in vivo calcium imaging. In comparison, lens design-based correction can enable even faster frame rates. Nevertheless, the volumetric correction with great

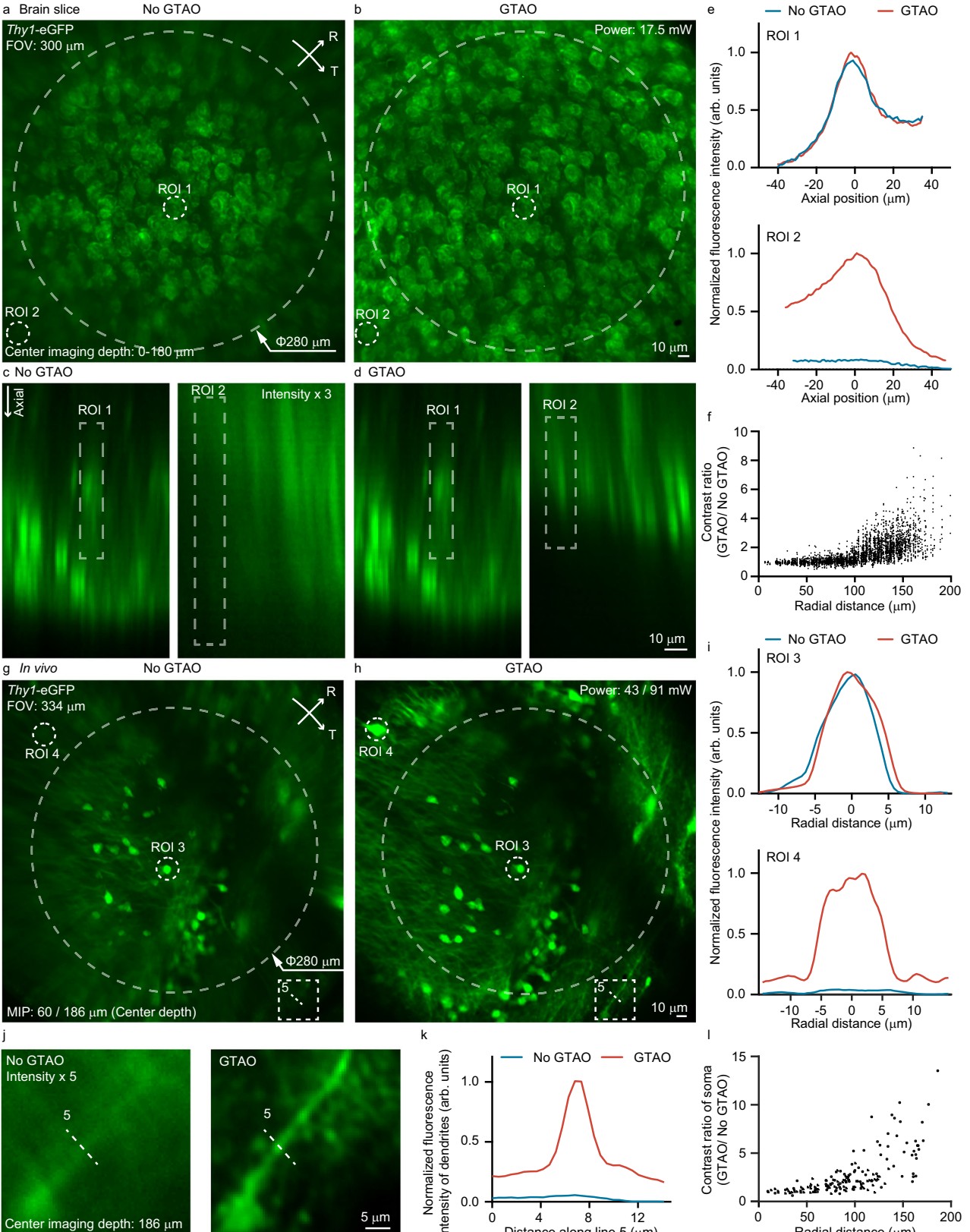

**Fig. 3 | GTAO-based structural imaging. a**, **b** Maximum intensity projection (MIP) of the image stack of *Thy1*-eGFP mouse brain slice along the axial direction acquired without and with GTAO, respectively. **c**, **d** The axial cross-sectional images of two regions of interest (ROI) without and with GTAO, respectively. **e** The axial plot of ROI 1 and 2. **f** The contrast ratio over the imaging FOV (*n* = 3017 ROIs from one brain slice). The imaging experiment was repeated independently 5 times with similar results. **g**, **h** MIP of in vivo structural images of *Thy1*-eGFP mouse brain without and with GTAO, respectively. **i** Cross-sectional plot for ROI 3 and 4. **j** Zoomed-in view of ROI 5 (dendrites). **k** Cross-sectional plot along the dashed line in ROI 5. **l** Image contrast ratio over the FOV (*n* = 111 somata from 2 mice). The in vivo structural imaging was repeated independently in 5 mice with similar results.

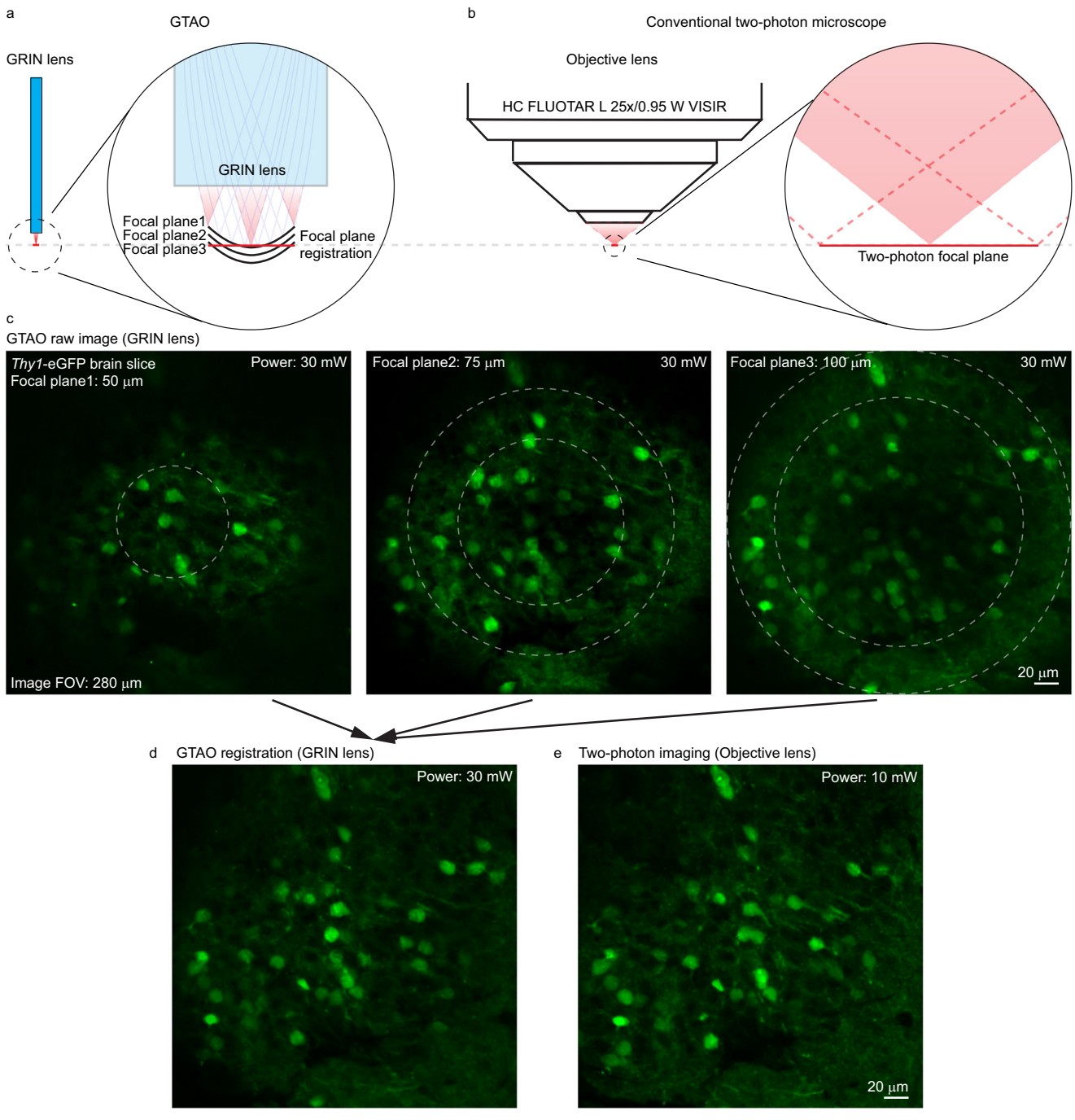

**Fig. 4 | 3D image registration. a, b** Focal plane through GRIN lens with GTAO and through an objective lens, respectively. **c** Two-photon image stack of *Thy1*-eGFP brain slice through GRIN lens. **d** Two-photon image through GRIN lens after image processing. **e** The same image plane recorded through the objective lens. The *Thy1*-eGFP registration imaging was repeated independently 3 times with similar results.

adaptability in the choices of GRIN lenses and wavelength makes GTAO a powerful and versatile solution for two-photon deep tissue functional imaging applications.

## Methods
### Animal
The research work complied with all relevant ethical regulations. All procedures involving mice were approved by the Purdue University Animal Care and Use Committee (protocol number: 1506001267). Wild-type (WT) C57BL/6 mice for virus injection and *Thy1*-eGFP M line mice were obtained from the Jackson Laboratory. The viruses (AAV1-*Syn*-GCaMP6s-WPRE-SV40) were purchased from Addgene. The mice

were housed in the animal facility of the Bindley Bioscience Center at Purdue University. Surgical procedures were performed on adult male and female mice with the ages of two to four months.

### Optical alignment
For new installation or the replacement of the correction GRIN lens to better match the pitch number of the implanted GRIN lens, we need to adjust the orientation and position of GRIN lens 1. Its orientation needs to be along the optical axis and its position needs to be centered around the optical axis. During the alignment, we removed the objective lens (Obj. 1) in front of GRIN lens 1 such that a collimated on-axis laser beam was incident on the lens facet. We observed the

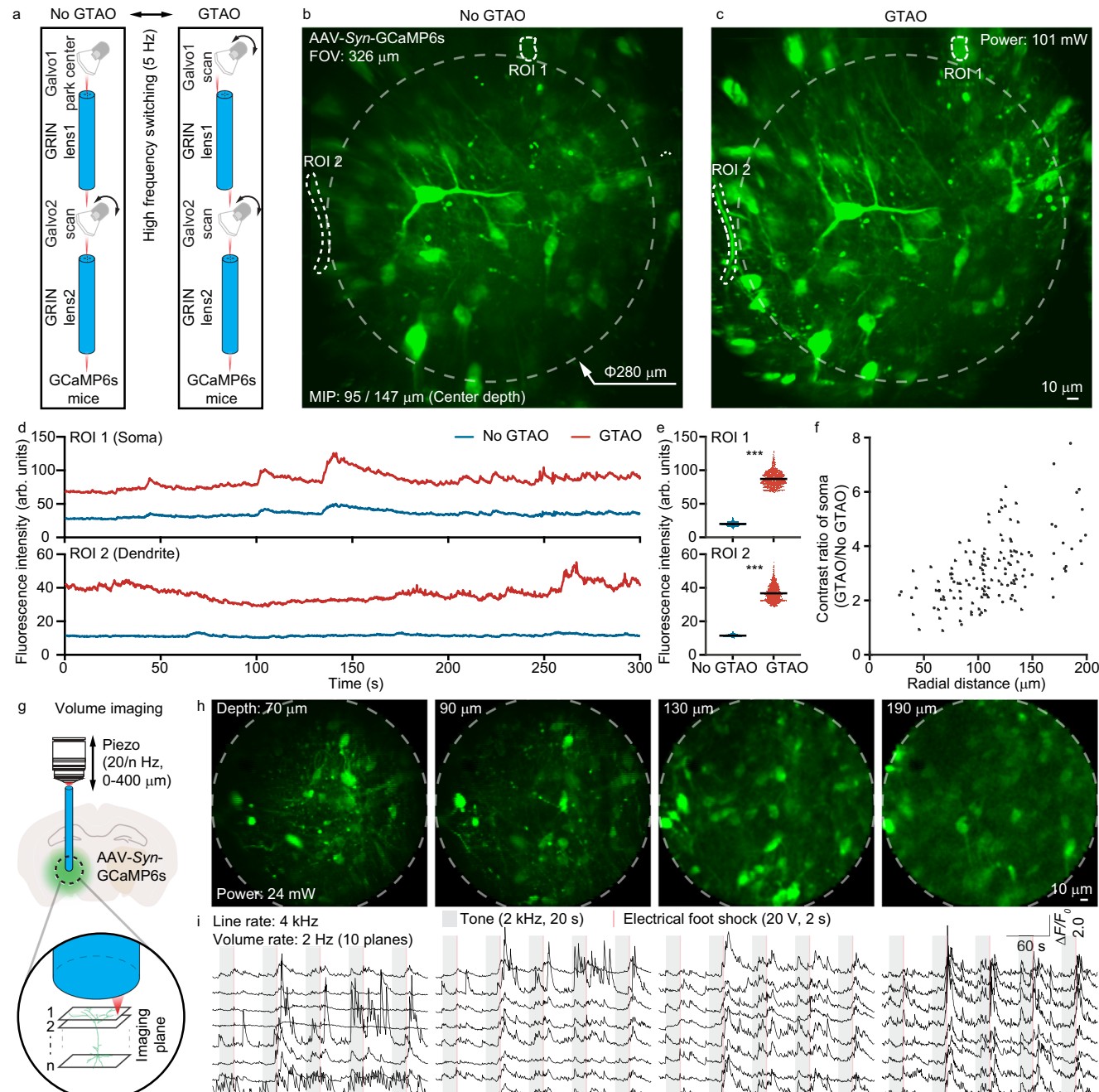

**Fig. 5 | GTAO-based functional imaging. a** The galvo control for near-simultaneous comparison of calcium imaging without and with GTAO. **b**, **c** Two-photon calcium imaging without and with GTAO, respectively. (d) Calcium transients from ROI 1 and 2. **e** Statistics of the calcium transients in d. Lines and error bars represent the mean ± standard error of the mean. The scattered dot plot represents individual data values. Two-sided Wilcoxon matched-pairs signed rank test, ***$P = 1.0089e\text{-}246$, $n = 1500$ frames for ROI 1 and 2. **f** The ratio of the image contrast, $n = 141$ somata from 4 mice. **g** Configuration for GTAO-based volumetric calcium recording. **h** Calcium images at 70, 90, 130, and 190 μm depth. **i** The corresponding calcium transients from these four image planes in response to external stimulation. The in vivo calcium imaging was repeated independently in 6 mice with similar results.

reflection by the GRIN lens facet and adjusted the lens orientation such that its reflection counter-propagated with respect to the incident laser beam. After the orientation adjustment, we put the objective lens back. We then added a beam splitter (at position A1 in Supplementary Fig. 8) to observe the GRIN lens facet through the objective lens. We controlled Galvo 1 to position the laser focus on four spots that resided on a 500 μm diameter circle centered in the middle of the FOV. We translated GRIN lens 1 such that the four spots were in focus and right on the edge of the GRIN lens. In the final step, we adjusted the position of the second objective lens (Obj. 2) such that the on-axis laser beam remained on-axis and the beam was collimated. For the second GRIN

lens under the imaging objective lens, we need to perform alignment frequently. To simplify the process, we used a 532 nm laser diode to illuminate the GRIN lens facet at an angle and monitor its reflection on a camera (Supplementary Fig. 8). For in vivo studies, the animal brain was mounted on a tip-tilt stage. We adjusted the stage such that the reflected laser spot was centered on the target location. To center the GRIN lenses in the middle of the laser scanning field, we added a beam splitter using a magnetic mount (at position A2 in Supplementary Fig. 8), which could be conveniently removed. Through the beam splitter, we used a camera to observe the GRIN lens facet. We commanded the laser scanners to dwell on four spots that resided on a

500 μm diameter circle centered in the middle of the scanning FOV. We translated the GRIN lens such that all four spots were in focus and on the edge of the GRIN lens facet.

## GRIN lens calibration

The aberration correction of GTAO is achieved by the scanning of laser beams through the two GRIN lenses. A key parameter for a GRIN lens is its physical length per pitch. This parameter is directly linked with the radial refractive index profile and is also associated with the maximum NA at a given lens diameter. With different pitch lengths, the amount of astigmatism at the same field position will be different. To accommodate the variation of GRIN lenses (e.g. different lens makers, batch), we need to perform calibration to the scanner driving signal. Experimentally, we positioned the second GRIN lens under the imaging objective lens and then positioned an observation objective lens and a camera under the GRIN lens to view the transmitted laser spot. Firstly, we used the first scanner to move the laser spot to e.g. 140 μm away from the axis. Secondly, we used the second scanner to undo this shift and move the spot along the orthogonal direction until the laser spot appeared round and symmetrical and the peak intensity was maximized. Through this process, we determined the driving signal voltage range for both scanners. After the static correction, we drove both scanners at the typical imaging speed (e.g. 4 kHz line rate). We would then fine-tune the amplitude and phase of the driving signals to achieve optimal correction, which completed the calibration process.

## Fear conditioning

For the in vivo volumetric calcium imaging, we employed fear conditioning to elicit calcium responses in mice. The mouse's head was secured under a microscope, while its four limbs were positioned on two separate metallic plates made of tin foil. These foil sheets were electrically isolated from each other. Each foil plate was connected to an electrode of the stimulator. Each stimulation cycle lasted for one minute. It started with a 10-second baseline period, followed by a 20-second tone stimulus at 2 kHz and another 30-second baseline period. At the final 2 seconds of the tone stimulus, a 20 V foot shock was administered to the mouse. Each mouse underwent a minimum of five stimulation cycles.

## Data analysis

All the time-lapse images were spatially registered with the averaged image through the utilization of mutual information, ensuring accurate motion correction. Relative fluctuations in fluorescence intensity ($\Delta F/F_0$) were precisely quantified from designated regions of interest (ROI). $F_0$ was defined as the minimum 10% of the fluorescence signal within each ROI.

## Statistics

All the data in this study are represented as mean ± standard error of the mean or scatterplot. The Shapiro–Wilk normality test was performed for all the data. The Wilcoxon matched-pairs signed rank test was selected for comparison in Fig. 5e. $P < 0.05$ is recognized as statistically significant. All statistical analyses were performed using GraphPad Prism (10.1.0), and MATLAB (2021a). The P-value of Fig. 5e is calculated using the "signrank" function in the Statistics and Machine Learning Toolbox of MATLAB software. No results of the successful acquisition from images and measurements were excluded and filtered. The experiment did not include randomization and blinding. n, P values, and the statistical tests were shown in the figure legends.

## Reporting summary

Further information on research design is available in the Nature Portfolio Reporting Summary linked to this article.

## Data availability

The data generated in this study have been deposited in the Science Data Bank database[50] [https://doi.org/10.57760/sciencedb.13798].

## Code availability

The code used in this study is available in the Science Data Bank [https://doi.org/10.57760/sciencedb.13796].

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

## Acknowledgements

This work was supported by NIH grants R21EY032382 (M.C.), U01NS126054 (M.C.), and U01NS118302 (M.C.). The funders had no role in study design, data collection and analysis, the decision to publish, or the preparation of the manuscript. M.C. thanks Howard Hughes Medical Institute for scientific instruments.

## Author contributions

M.C. invented GTAO and supervised the project. Y.L. and M.C. designed the GTAO-based two-photon imaging system. Y.L. implemented the overall imaging system. C.W., J.L., and H.J. assisted in the construction of the imaging system and the calibration procedure. Y.L. and Z.C. collaborated on the experiment, data analysis, and figure preparation. Y.L. and J.L. measured the objective lens dispersion. M.C. wrote the manuscript with input from all authors.

## Competing interests

Purdue Research Foundation filed a patent (US20230333368A1, Pending) for the GTAO-based two-photon imaging system, which covered the concept, design, and implementation. M.C. was the inventor. The remaining authors declare no competing interests.
