## [Peer Review File · Nature Communications]

Geometric transformation adaptive optics (GTAO) for volumetric deep brain imaging through gradient-index lensesReviewer #1 (Remarks to the Author):

GRIN microendoscopy stands as a gold-standard technique for minimally invasive access to deep brain structures. However, it fundamentally suffers from spatially inhomogeneous high-order aberrations, particularly fourth-order astigmatism. Due to its quadratic dependence on the radial axis of the GRIN lens, astigmatism largely constrains the field-of-view, thereby reducing the number of neurons to be investigated. The widely recognized solution is wavefront shaping but compensating fourth-order astigmatism is practically challenging, particularly within biological samples prone to physiological motion. This manuscript presents an interesting solution to this technical issue by employing the GRIN lens itself for the pre-compensation of fourth-order astigmatism. Attached are my specific comments.

1. As depicted in Fig. 1a, the GTAo in principle can correct astigmatism, even at the periphery of the GRIN lens. Nonetheless, the provided image data is confined to a limited field-of-view (e.g., 280 μm in Fig. 1d&1e). How is this effective FOV determined? It would be informative to incorporate complete field-of-view data, at least in Supplementary Figures, and also elaborate on the theoretical basis for the imposed limitation on the effective field-of-view.
2. While the spatial light modulator is integrated into the setup to offer static aberration correction, a comprehensive characterization of this component is absent. but its detailed characterization is missing. It would be valuable to know whether a comparable enhancement can be achieved when the static phase correction is absent.
3. As mentioned in the Discussion section, the GRIN lenses are not made equal owing to fabrication errors. While this concern is acknowledged and somewhat addressed through the experiment involving different GRIN lenses (Fig. 2f), the accompanying description remains somewhat qualitative. The underlying theoretical basis needs to be clarified.
4. It would be beneficial to explore the potential drawbacks of GTAo. While it evidently aids in astigmatism reduction, there could be a trade-off in terms of elevated other aberrations. Additionally, it's important to address concerns such as the impact on group delay dispersion (GDD) and potential power loss. These practical considerations warrant clarification.
5. Considering the user's standpoint, the possibility of employing various GRIN lenses with differences in pitch length, diameter, or working distance raises questions about the ease of realigning the optical system when switching to a different GRIN lens. Elaborating on this matter would be valuable for readers contemplating the adoption of this system.

Reviewer #2 (Remarks to the Author):

This work by Li et al. presents aberration correction, in particular astigmatism correction, via cascade of GRIN lens. The proposed GTAo technique is based on a 90-degree rotation of the beam spatial profile between the two GRIN lenses. The authors developed a smart dual scanner scheme to accommodate the gradual increase of astigmatism from the GRIN lens center to the outer region, enabling large field-of-view mapping. The basic optical principle and the volumetric imaging capacity were nicely demonstrated by fluorescent beads in Fig. 1 and 2. I am impressed that GTAo is adaptable to the variation of GRIN lenses and offers inherent achromatic astigmatism. These are the questions I had in my mind when I read the abstract. The authors are apparently well experienced in this field, so they prepared convincing results to address these potential concerns. Furthermore, to attract audience from the neuroscience community, which meets the interdisciplinary requirement of Nat. Comm., the authors performed in vitro and in vivo observations of mouse brains, manifesting that the aberration correction was real-time during volumetric recording.

Overall speaking, the idea is elegant, and the results are convincing. Nevertheless, I do have concerns on the novelty and practical applicability of this work. First, the idea of GRIN lens cascade

has been proposed by Martin Booth, who also presented improvement of axial resolution (Nat. Comm. 10, 4264 (2019)). Second, the AO correction of GRIN lens imaging has been demonstrated in many prior arts. In particular, ref. 42 also shows large volumetric imaging with AO correction. The volume size in ref. 42 is 300 μm in diameter, and 300 μm in depth, both similar to this work, but the setup in ref. 42 is much simpler. The complicated setup in this manuscript might be the main bottleneck when promoting the idea to neuroscience labs. Therefore, while I have no doubt that this work is publishable as an innovative optical design (two scanner + two GRIN lens) in a specialized journal for optics, I am afraid that I cannot recommend its publication in a high-profile interdisciplinary journal like Nat. Comm.

Following are specific concerns:

1. The main statement of the paper is the presence of field dependent astigmatism in GRIN lenses. The paper then describes a technique to solve this problem. In my opinion the motivation of the paper is poorly documented and the motivation also lacks a description of the impact. The technical reason of the presence of astigmatism in GRIN lenses is not explained and I do not think it is obvious. Please add an explanation about it. Ref 29 is a generic book of optics and it is not enough. Intuitively, I think this is a problem that increases for high NA, large FOV, GRIN lenses. If this is the case, to estimate the impact of this paper it is important to add information about the importance for neuroscience imaging of using these GRIN lenses.
2. From the technical point of view some improvements about the presentation must be carried out. The most common description of aberrations is carried out often using Zernike polynomials. In this paper the authors use Seidel polynomials that appear to be a convenient choice because of the presence of field dependent astigmatism. Please explain this in the text.
3. It is well known that there are different aberrations in GRIN lens. For example, in Fig. 2 of Ref. 43, the contribution of spherical aberration is the same as astigmatism. Furthermore, coma is also one of the main aberration sources. The authors need to explain whether their design compensates for other aberrations, or works only for astigmatism.
4. Please evaluate how much aberration is contributed by the inherent birefringence of the GRIN lens (He et al., Nat. Comm. 10, 4264 (2019)), and comment on whether this dual GRIN lens design is able to correct the intrinsic polarization/phase aberrations.
5. The volumetric imaging results are impressive. However, when performing volumetric imaging under a GRIN lens, the magnification of each layer is often different. A common situation is that as the focus moves up, the image undergoes contraction around the FOV center, i.e. image distortion in the axial direction. The authors shall specify whether the results have been post-processed, or whether the dual GRIN lens design is able to tackle the axial distortion.
6. In Fig. 1a, the authors mentioned "astigmatism ...through the first GRIN lens reached 4.6 waves from the peak to the valley." It is more informative if the authors provide Zernike or Seidel order analysis, in order to confirm that the 4.6 waves aberration is mainly caused by astigmatism.
7. The specifications of most optical components are missing, e.g. objectives, SLM, galvo, etc. Note that in p. 6, the sentence "positioned a water immersion observation objective lens (Olympus 60x NA 1.2) under the GRIN lens" causes the confusion of whether the NAs of the objectives and the GRIN lenses match each other.
8. In Fig. 3b, the neurons outside the 280- μm diameter are clearly visible, but in Fig. 3h, the outside region disappears. What happened to the region outside the 280- μm diameter?
9. In p. 9, "we performed a digital pixel shifting in the axial direction (Fig. 4c, d)." Please explain the detail of the "digital pixel shifting" process, which might link to my comment 3.
10. Given that a series of optical sections of a pyramidal cell is illustrated in Fig. 5g, it would be more persuasive if 3D reconstructed dendritic structures of neurons were presented.
11. P.3, line 3, prims  prism

Dear Reviewers,

Thank you very much for the very thoughtful reading of our manuscript, “Geometric transformation adaptive optics (GTAO) for volumetric deep brain imaging through gradient-index lenses”, and the many insightful comments. We have revised our manuscript accordingly to provide new information, data, and discussion. Below, we address every comment in detail.

REVIEWER COMMENTS

Reviewer 1:

GRIN microendoscopy stands as a gold-standard technique for minimally invasive access to deep brain structures. However, it fundamentally suffers from spatially inhomogeneous high-order aberrations, particularly fourth-order astigmatism. Due to its quadratic dependence on the radial axis of the GRIN lens, astigmatism largely constrains the field-of-view, thereby reducing the number of neurons to be investigated. The widely recognized solution is wavefront shaping but compensating fourth-order astigmatism is practically challenging, particularly within biological samples prone to physiological motion. This manuscript presents an interesting solution to this technical issue by employing the GRIN lens itself for the pre-compensation of fourth-order astigmatism. Attached are my specific comments.

We thank the reviewer for the insightful summary. The GRIN lens-based deep mouse brain imaging is very limited in imaging field-of-view (FOV) because the aberration of the lens system is dominated by fourth-order astigmatism which is a quadratic function of field position. If the goal is to image static structure through the GRIN lenses, conventional AO solutions will suffice. However, for imaging fast dynamics, none of the reported solutions meet the practical requirements.

The key application in neuroscience is calcium imaging, which demands near-simultaneous *in vivo* imaging of neuronal dynamics over large FOV at high throughput. Thus, the ideal AO correction solution should provide spatially continuously varying AO corrections that can keep up with the raster scanning. So far, none of the established methods can provide a satisfactory solution to this challenge. The common static pupil-plane-based AO can provide a very limited correction FOV. Thus, for large FOV recordings, the wavefront needs to be updated for each sub-region of the FOV. In our lab’s earlier study, a total of 52 small FOV images corrected by 52 wavefronts were stitched to cover the entire FOV. In addition to the much-reduced imaging throughput, a more severe problem is the strong motion artifact. In conventional raster-scanning-based imaging, the adjacent pixels and lines are continuous in time. Thus, the image distortion appears smooth and continuous. With the sequential sub-image recording, there was temporal discontinuity between adjacent sub-images. Therefore, the motion could induce discontinuity in the images. A more practical solution for high-throughput large FOV calcium imaging is through lens design. However, the reported solution is limited in the length of the GRIN lens (not long and slim enough for deep brain imaging) and the correction is not volumetric (the AO correction is valid for a specific working distance). Moreover, the solution lacks adaptability.

In this work, we reported GTAO which enables continuous volumetric AO correction over large FOV for high-throughput dynamic imaging. The correction is not only volumetric but also inherently achromatic. An important feature of GTAO is its adaptability. GRIN lenses of various pitch lengths, makers, and batches can all be corrected through the same setup. We demonstrated *in vivo* large FOV volumetric imaging in mouse brain.

1. As depicted in Fig. 1a, the GTAO in principle can correct astigmatism, even at the periphery of the GRIN lens. Nonetheless, the provided image data is confined to a limited field-of-view (e.g., 280 μm in Fig. 1d&1e). How is this effective FOV determined? It would be informative to incorporate complete field-of-view data, at

least in Supplementary Figures, and also elaborate on the theoretical basis for the imposed limitation on the effective field-of-view.

We thank the reviewer for the insightful comment. Through bead imaging, we quantified the point spread function (PSF) variation over the FOV. At ~ 280 -micron FOV, the peak intensity of the fluorescence intensity dropped to $\sim 50\%$ of the value in the middle of the FOV. Thus, we controlled our EO modulator during the imaging to turn off the laser power for regions outside this circle. We appreciate the inspiring comment from the reviewer. For imaging neurons, the soma is much greater than the beads. Therefore, even at larger FOV, we can still nicely resolve the soma. So, for soma resolution recording, the usable FOV can indeed exceed the 280-micron circle as shown in the tissue slice imaging data. To make this point clear to the reader, we have performed new imaging experiments, which showed even larger FOV (e.g. 334×334 micron²) for *in vivo* structural and calcium imaging. We have updated Fig. 3, 5, and Video 2, 3 in the manuscript accordingly.

As an example, we show the *in vivo* image comparison in the updated Fig. 3. The new scanning FOV has been increased to 334 microns. The diagonal size of the image reached 472 microns, approaching the physical diameter of the GRIN lens (500 microns). The new measurement suggests that GTA0 (right) can nicely resolve single neurons and even dendrites near the boundary of the 334-micron FOV. In comparison, the resolvable cells were confined to the middle of the FOV without GTA0 (left).

2. While the spatial light modulator is integrated into the setup to offer static aberration correction, a comprehensive characterization of this component is absent. but its detailed characterization is missing. It would be valuable to know whether a comparable enhancement can be achieved when the static phase correction is absent.

We thank the reviewer for the insightful question. The SLM was used to remove the invariant static on-axis (in the middle of the FOV) aberration of the entire imaging system, which includes the aberration of the galvo scanners and the on-axis aberration of all the optical components. To evaluate the SLM's contribution, we performed new experiments and recorded images with the SLM correction enabled and then with the SLM correction disabled (flat wavefront after the SLM). We added supplementary Fig. 5 to show the new data. The experimental results show that the SLM correction was useful and made the central image sharper and of higher brightness. However, the SLM correction made no contribution to extending the FOV (upper left and lower left

images). In comparison, GTA0 greatly extended the FOV (upper right and lower right images). In particular, even without SLM correction, the neurons and dendrites in the lower region of the image (encircled by orange dashed lines) were visible with GTA0. However, without GTA0, they appeared very blurry even with the help of the SLM.

Thy1-YFP (*in vivo*) No GTA0

GTA0

3. As mentioned in the Discussion section, the GRIN lenses are not made equal owing to fabrication errors. While this concern is acknowledged and somewhat addressed through the experiment involving different GRIN lenses (Fig. 2f), the accompanying description remains somewhat qualitative. The underlying theoretical basis needs to be clarified.

The GRIN lens from different makers or batches may have different pitch lengths (the physical length of the 1-pitch GRIN lens), which is a key parameter related to the index profile and the NA of the GRIN lens. As a

result, the amount of astigmatism at the same imaging FOV may vary. However, the overall fourth-order astigmatism remains a quadratic function of field position. Therefore, by scaling the scanning range on the first GRIN lens (the correction GRIN lens), we could accommodate such variations. To make this process clear to the reader, we added the following paragraph to the method section (GRIN lens calibration) of the manuscript.

“A key parameter for a GRIN lens is its physical length per pitch. This parameter is directly linked with the radial refractive index profile and is also associated with the maximum NA at a given lens diameter. With different pitch lengths, the amount of astigmatism at the same field position will be different.”

In addition, we also revised Supplementary Fig. 4b-d to show the aberration difference of two lenses at the same field position. As astigmatism has a quadratic dependence on the field position, adjusting the ratio of the scanning range of the two scanners can balance the amount of astigmatism from the two different GRIN lenses over the entire FOV. For example, the difference in pitch length (4.96 vs. 5.67 mm) leads to the NA difference (NA 0.5 vs. NA 0.44) and the radial index profile difference (shown in d). As a result, the amount of astigmatism (shown in e) is different. However, as both are quadratic functions of radial position, we just need to scale the radial position to compensate for the magnitude difference. For example, by setting their radial position ratio as 1.07:1 between the NA 0.44 and the NA 0.5 GRIN lens, the astigmatism from the two lenses can be precisely matched over the entire FOV.

4. It would be beneficial to explore the potential drawbacks of GTA0. While it evidently aids in astigmatism reduction, there could be a trade-off in terms of elevated other aberrations. Additionally, it's important to address concerns such as the impact on group delay dispersion (GDD) and potential power loss. These practical considerations warrant clarification.

We thank the reviewer for the insightful comment. The aberration of the GRIN lens is dominated by fourth-order astigmatism. With GTA0, the astigmatism was canceled but there was surely residual aberration as shown in Fig. 1a. Nevertheless, the advantage of canceling astigmatism outweighs the residual aberration, as shown in the greatly extended FOV (image information content) during *in vivo* volumetric recording. To make the information clear to the readers, we added supplementary Fig. 2, in which we analyzed the GRIN lens aberration (with radial position = 140 microns), showed the scale of different aberration terms (Zernike coefficients), and compared the Strehl ratio of a focus with the astigmatism, spherical aberration, and coma, individually. The analysis showed that astigmatism is the dominant term and the residual aberration is far less in comparison. In addition, we also added information about the overall system GDD and the power throughput to the last paragraph on page 6 of the manuscript. The power throughput was 25% and the overall GDD was

~12,000 fs², which was within the range of the built-in pulse compressor in femtosecond lasers and commercial prism compressors.

Zernike Coefficients Uncorrected	Zernike Coefficients Corrected	j	Zernike Polynomial $Z_j(\rho, \theta)$	Aberration
-0.00001187	-0.01034903	5	$\sqrt{6}\rho^2 \sin(2\theta)$	Primary astigmatism at 45°
-0.95650143	0.01225403	6	$\sqrt{6}\rho^2 \cos(2\theta)$	Primary astigmatism at 0°
0.00000219	-0.02052655	7	$\sqrt{8}(3\rho^3 - 2\rho)\sin\theta$	Primary y coma
0.01306444	0.00724213	8	$\sqrt{8}(3\rho^3 - 2\rho)\cos\theta$	Primary x coma
-0.00000069	-0.06563088	9	$\sqrt{8}\rho^3 \sin(3\theta)$	
-0.06711070	-0.06626744	10	$\sqrt{8}\rho^3 \cos(3\theta)$	
-0.03759695	-0.04735100	11	$\sqrt{5}(6\rho^4 - 6\rho^2 + 1)$	Primary spherical
0.02225823	0.00264508	12	$\sqrt{10}(4\rho^4 - 3\rho^2) \cos(2\theta)$	Secondary astigmatism at 0°
0.00000021	-0.00232457	13	$\sqrt{10}(4\rho^4 - 3\rho^2) \sin(2\theta)$	Secondary astigmatism at 45°
-0.00172994	0.00103271	14	$\sqrt{10}\rho^4 \cos(4\theta)$	
-0.00000004	-0.01470172	15	$\sqrt{10}\rho^4 \sin(4\theta)$	
-0.00153786	-0.00197576	16	$\sqrt{12}(10\rho^5 - 12\rho^3 + 3\rho)\cos\theta$	Secondary x coma
0.00000028	0.00003305	17	$\sqrt{12}(10\rho^5 - 12\rho^3 + 3\rho)\sin\theta$	Secondary y coma
0.00146188	0.00186694	18	$\sqrt{12}(5\rho^5 - 4\rho^3) \cos(3\theta)$	
0.00000002	0.00153084	19	$\sqrt{12}(5\rho^5 - 4\rho^3) \sin(3\theta)$	
-0.00007174	0.00073927	20	$\sqrt{12}\rho^5 \cos(5\theta)$	
-0.00000001	-0.00157862	21	$\sqrt{12}\rho^5 \sin(5\theta)$	
-0.00043696	-0.00024351	22	$\sqrt{7}(20\rho^6 - 30\rho^4 + 12\rho^2 - 1)$	Secondary spherical

5. Considering the user's standpoint, the possibility of employing various GRIN lenses with differences in pitch length, diameter, or working distance raises questions about the ease of realigning the optical system when switching to a different GRIN lens. Elaborating on this matter would be valuable for readers contemplating the adoption of this system.

We thank the reviewer for the helpful comment. Compared with other correction methods, a key advantage of GTA0 is its adaptability to the variation of GRIN lenses. To facilitate the adoption of GTA0 in the new user's lab, we added more detailed information to Optical Alignment in the method section, which contains step-by-step information about the process. In summary, for slight pitch lens variation (e.g. different batches, or the same 1.5 pitch from different vendors), we do not need to change the optical components. Instead, we just need

to rescale the scanning range of Galvo 1. For major length changes (e.g. changing from 1.5 pitch to 2 pitch), we will need to replace GRIN lens 1. The key considerations are that the orientation of the GRIN lens needs to be along the optical axis and its position needs to be centered around the optical axis. First, we removed the objective lens (Obj. 1) in front of GRIN lens 1. Thus, a collimated on-axis laser beam was incident on the facet of GRIN lens 1. We observed the reflection from the facet and adjusted the orientation of GRIN lens 1 such that the reflected beam counter propagated with respect to the incident beam. After the orientation adjustment, we put the objective lens back. We then added the beam splitter (at position A1 in Supplementary Fig. 8) to observe the GRIN lens facet through the objective lens. We controlled Galvo 1 to position the laser on four spots at equal distances to the center of the FOV. We translated GRIN lens 1 such that the four spots were in focus and right on the edge of the GRIN lens. In the final step, we adjusted the position of the second objective lens (Obj. 2) such that the on-axis laser beam remained on axis and the beam was collimated.

Reviewer 2:

This work by Li et al. presents aberration correction, in particular astigmatism correction, via cascade of GRIN lens. The proposed GTAo technique is based on a 90-degree rotation of the beam spatial profile between the two GRIN lenses. The authors developed a smart dual scanner scheme to accommodate the gradual increase of astigmatism from the GRIN lens center to the outer region, enabling large field-of-view mapping. The basic optical principle and the volumetric imaging capacity were nicely demonstrated by fluorescent beads in Fig. 1 and 2. I am impressed that GTAo is adaptable to the variation of GRIN lenses and offers inherent achromatic astigmatism. These are the questions I had in my mind when I read the abstract. The authors are apparently well experienced in this field, so they prepared convincing results to address these potential concerns. Furthermore, to attract audience from the neuroscience community, which meets the interdisciplinary requirement of Nat. Comm., the authors performed *in vitro* and *in vivo* observations of mouse brains, manifesting that the aberration correction was real-time during volumetric recording.

We thank the reviewer for the insightful summary of the GTAo technology. GRIN lens is currently the widely used solution to access deep brain and inner tissue. However, a key challenge of using GRIN lenses (e.g. 0.5 mm in diameter, 6-7 mm long) is the small FOV, which is due to the fourth-order astigmatism (a quadratic function of field position). Due to its spatially varying nature, fourth-order astigmatism is difficult to correct for *in vivo* dynamic functional imaging applications. If the goal is just to image static structures, conventional pupil plane AO developed over the past decades can easily handle that. But for neuroscience and many other biomedical applications, the key to *in vivo* measurements is to obtain near-simultaneous dynamic information over large FOV at high throughput. Thus, the correction methods need to keep up with the high-speed scanning and can robustly handle tissue motion. Currently, none of the reported solutions can provide a satisfactory solution.

Take the conventional pupil-plane AO as an example, each correction at the pupil plane is only valid for a small sub-FOV. Thus, many wavefronts and sub-images are required to synthesize the full FOV. In our lab's early studies, we explored such a solution, in which we were able to switch the pupil plane wavefront at ~140 microseconds and employed a total of 52 wavefronts to image 52 sub-images which can be stitched to form a large FOV which matches the FOV of GTA0. For *in vivo* applications, such methods have two main problems. One is the reduced imaging throughput. The most important application in neuroscience is calcium imaging, which demands near-simultaneous large FOV coverage. The other is the very severe motion artifacts. With the conventional full FOV raster scanning (as in the application of GTA0), the adjacent lines and pixels are continuous in time. Therefore, the issue motion will distort the image in a spatially smooth and continuous manner. With the conventional pupil-plane AO-based sub-image recording, the neighboring region between these many sub-images is discontinuous in time, leading to discontinued images during tissue motion. For these reasons, none of the pupil-plane AO methods can match GTA0 in FOV coverage and imaging throughput for practical *in vivo* calcium imaging applications.

Overall, different from conventional pupil-plane AO, GTA0 provides an adaptable spatially continuous variation of the correction to precisely match the spatially varying fourth-order astigmatism.

Overall speaking, the idea is elegant, and the results are convincing. Nevertheless, I do have concerns on the novelty and practical applicability of this work. First, the idea of GRIN lens cascade has been proposed by Martin Booth, who also presented improvement of axial resolution (Nat. Comm. 10, 4264 (2019)).

We thank the reviewer for the helpful comment. Although the beautiful paper by Prof. Booth involves using multiple units of GRIN lenses, its essence is about using the very long GRIN lenses as polarizing components to control the polarization of light and there is nothing about the geometric transformation employed in their work or enabling large FOV imaging capability. In addition, the focal profile control work was right in the center of the FOV, which was inherently free from astigmatism. In comparison, GTA0 as the name suggests is about the rotation of the spatial profile of light, rather than exploiting the polarization. In other words, the polarization control paper does not achieve the large FOV aberration correction needed for *in vivo* calcium imaging. Instead, it explores the birefringence effect in the GRIN lens for polarization control. One key factor we hope to point out is that the birefringence in the GRIN lens is a very weak effect. As discussed in the 2019 paper, even at the visible optical wavelength, the birefringence through GRIN lenses is only 10^{-5} . One would need a 100 mm long lens to generate a micron optical path length difference. In comparison, the depth of the mouse brain is a bit above 6 mm and the typical GRIN lens for *in vivo* calcium imaging is only 6-7 mm in length. Moreover, the laser wavelength is ~920-930 nm, which has even lower birefringence than the 10^{-5} at the visible wavelength reported in the polarization control paper. In addition, the longer wavelength will demand even large optical path length differences to experience any polarization effect. Overall, the birefringence effect is not significant to the *in vivo* mouse brain calcium imaging applications.

To make this point very clear to the readers, we performed a new measurement and added a new Supplementary Fig. 1. Experimentally, we used a linearly polarized light as the input to the GRIN lens and observed the aberrated optical focus through the GRIN lens. If the focal profile distortion was due to the birefringent, the transmitted profiles should lack rotational symmetry due to the use of linearly polarized light (polarization along one direction). Instead, if the distortion was mainly due to astigmatism, the beam profiles should be rotationally symmetric. As expected, the data (both the tangential and sagittal focal planes) showed that the transmitted optical profile is indeed isotropic (rotationally symmetric), and free from any symmetry along the polarization direction. In addition, GTA0 explores no control on the laser polarization, and yet its correction can yield round focus across the entire imaging FOV (e.g. Fig. 1-3).

Second, the AO correction of GRIN lens imaging has been demonstrated in many prior arts. In particular, ref. 42 also shows large volumetric imaging with AO correction. The volume size in ref. 42 is 300 μm in diameter, and 300 μm in depth, both similar to this work, but the setup in ref. 42 is much simpler. The complicated setup in this manuscript might be the main bottleneck when promoting the idea to neuroscience labs. Therefore, while I have no doubt that this work is publishable as an innovative optical design (two scanner + two GRIN lens) in a specialized journal for optics, I am afraid that I cannot recommend its publication in a high-profile interdisciplinary journal like Nat. Comm.

We thank the reviewer for the helpful comment. As discussed previously, if the goal is to image a static sample, conventional pupil-plane AO developed over the past decades can mostly work. But the most important application in neuroscience is calcium imaging, which is about capturing the dynamics nearly simultaneously over large FOV or volume. Currently, none of the reported methods can provide a satisfactory solution. Ref. 42 is the conventional pupil-plane AO method. For imaging through a GRIN lens, each correction only works for a very small sub-FOV due to the spatially varying pupil-plane aberration profiles. A large number of sub-images and wavefronts are needed to synthesize the full FOV. For example, the second paragraph on page 4 of the paper states the following. ***“Because the aberration vary from site to site, we performed AO correction in a number of subregions of 50 micron x 50 micron each separated by a distance of 30 micron laterally and stitched these subregions together to form a high-resolution image of large FOV”*** With the 30 micron step size, a total of ~100 images will be needed to cover the 300 micron x 300 micron FOV. As discussed previously, this will lead to much-reduced imaging throughput and image distortion during awake animal studies. As such, the pupil-plane AO-based GRIN lens correction cannot help the widely employed calcium imaging in neuroscience applications to extend the recording FOV and throughput. In addition, the GRIN lens employed in Ref. 42 was 1.4 mm in diameter and 3.7 mm long (a length-to-diameter ratio of 2.6). The 300-micron FOV is only ~20% of the lens diameter. In comparison, the GRIN lens used in this work is 0.5 mm in diameter and 6.6 mm long (a much greater aspect ratio of 13.2 for minimally invasive imaging through the entire depth of the mouse brain, with much more severe optical aberration). The GTAo’s 334-micron FOV is 67% of the GRIN lens diameter. In terms of area coverage, the pupil-plane AO method covers 4.6% of the facet area but the GTAo work covers 45% of the facet area while imaging through a much longer and thinner lens. Overall, the aberration handled by GTAo is greater due to the extremely slim form factor, and yet GTAo provides a much greater percentage of FOV coverage. It was worth noting that the 1.4 mm diameter lens was rarely used in practical neuroscience studies on the mouse brain as it is too large and can lead to severe brain damage. In comparison, the 0.5 mm lens employed in this work is widely adopted in neuroscience studies.

In terms of imaging throughput, the calcium imaging in Ref. 42 is performed only for a few sub-images. There was no full FOV calcium recording. For example, the data shown in Fig. 5 of Ref. 42 only contained three sub-images scanned at 5 Hz. Thus, the total throughput was only 30 micron x 30 micron x 15 Hz. In comparison, the calcium imaging throughput of GTAo was 330 micron x 330 micron x 20 Hz, which is 161 times greater than that in Ref. 42. As most deep brain studies in neuroscience involve high-throughput calcium recording which requires near simultaneous full FOV imaging, GTAo is the better choice for such tasks.

With the detailed information considered, we believe that the GTAo’s advance is significant and apparent, especially for the widely used calcium imaging applications in neuroscience, which demand minimally invasive slim GRIN lenses of large aspect ratio, high imaging throughput, and robustness during tissue motion. GTAo’s technical advances and potential for wide applications in neuroscience warrant consideration for this interdisciplinary journal.

Following are specific concerns:

1. The main statement of the paper is the presence of field dependent astigmatism in GRIN lenses. The paper then describes a technique to solve this problem. In my opinion the motivation of the paper is poorly documented and the motivation also lacks a description of the impact. The technical reason of the presence of astigmatism in GRIN lenses is not explained and I do not think it is obvious. Please add an explanation about it. Ref 29 is a generic book of optics and it is not enough. Intuitively, I think this is a problem that increases for high NA, large FOV, GRIN lenses. If this is the case, to estimate the impact of this paper it is important to add information about the importance for neuroscience imaging of using these GRIN lenses.

We thank the reviewer for the very helpful comments and advice. We agree that the motivation discussion and the significance discussion for the GTAO technology needs improvement, especially for readers who may not use GRIN lenses in their routine research and applications. Overall, *in vivo* imaging in the brain or other organs requires minimally invasive imaging devices. Thus the 0.5 mm diameter GRIN lenses of 6-7 mm in length hold great significance to many important biomedical applications. In particular, for neuroscience studies, the 0.5 mm lens is the gold standard. Currently, none of the established AO technology or lens corrector designs provide a satisfactory solution due to various problems (e.g. throughput, motion artifact, volumetric correction, and adaptability). For example, the pupil-plane AO cannot keep up with the variation of aberration across the imaging FOV during fast raster scanning and suffers from low imaging throughput and motion artifacts. The demonstrated aspheric corrector design worked with a relatively short GRIN lens (e.g. 0.5 mm diameter, 4 mm long) and the correction is not volumetric (the corrector is designed for a specific working distance, imaging above and below that plane still suffers from insufficient correction). In comparison, GTAO is well tailored for the *in vivo* volumetric recording applications, which fully meet the demand for *in vivo* calcium recording. We have revised the second paragraph in the manuscript to highlight the deficiency of all previous attempts and added the following.

“Overall, none of these explored solutions can meet all major practical requirements of in vivo volumetric functional imaging through the commonly used slim GRIN lenses (e.g. 0.5 mm in diameter, 7 mm long).”

Ref. 29 is an excellent book on optical lens design. To be more specific to the reference content, we revised the reference to the second chapter of this book, which discussed the imaging system aberration.

The dominant fourth-order astigmatism originates from the design of GRIN lenses. The ideal GRIN lens is designed for a perfect relay of on-axis spots, and it features a Hyperbolic Secant index distribution. The direct result of such a design is strong astigmatism (quadratic dependence of field position) for off-axis spots. We have performed a new analysis and showed the weight of different types of aberrations. We have added the new information to Supplementary Fig. 2.

Zernike Coefficients Uncorrected	Zernike Coefficients Corrected	j	Zernike Polynomial $Z_j(\rho, \theta)$	Aberration
-0.00001187	-0.01034903	5	$\sqrt{6}\rho^2 \sin(2\theta)$	Primary astigmatism at 45°
-0.95650143	0.01225403	6	$\sqrt{6}\rho^2 \cos(2\theta)$	Primary astigmatism at 0°
0.00000219	-0.02052655	7	$\sqrt{8}(3\rho^3 - 2\rho)\sin\theta$	Primary y coma
0.01306444	0.00724213	8	$\sqrt{8}(3\rho^3 - 2\rho)\cos\theta$	Primary x coma
-0.00000069	-0.06563088	9	$\sqrt{8}\rho^3 \sin(3\theta)$	
-0.06711070	-0.06626744	10	$\sqrt{8}\rho^3 \cos(3\theta)$	
-0.03759695	-0.04735100	11	$\sqrt{5}(6\rho^4 - 6\rho^2 + 1)$	Primary spherical
0.02225823	0.00264508	12	$\sqrt{10}(4\rho^4 - 3\rho^2) \cos(2\theta)$	Secondary astigmatism at 0°
0.00000021	-0.00232457	13	$\sqrt{10}(4\rho^4 - 3\rho^2) \sin(2\theta)$	Secondary astigmatism at 45°
-0.00172994	0.00103271	14	$\sqrt{10}\rho^4 \cos(4\theta)$	
-0.00000004	-0.01470172	15	$\sqrt{10}\rho^4 \sin(4\theta)$	
-0.00153786	-0.00197576	16	$\sqrt{12}(10\rho^5 - 12\rho^3 + 3\rho)\cos\theta$	Secondary x coma
0.00000028	0.00003305	17	$\sqrt{12}(10\rho^5 - 12\rho^3 + 3\rho)\sin\theta$	Secondary y coma
0.00146188	0.00186694	18	$\sqrt{12}(5\rho^5 - 4\rho^3) \cos(3\theta)$	
0.00000002	0.00153084	19	$\sqrt{12}(5\rho^5 - 4\rho^3) \sin(3\theta)$	
-0.00007174	0.00073927	20	$\sqrt{12}\rho^5 \cos(5\theta)$	
-0.00000001	-0.00157862	21	$\sqrt{12}\rho^5 \sin(5\theta)$	
-0.00043696	-0.00024351	22	$\sqrt{7}(20\rho^6 - 30\rho^4 + 12\rho^2 - 1)$	Secondary spherical

2. From the technical point of view some improvements about the presentation must be carried out. The most common description of aberrations is carried out often using Zernike polynomials. In this paper the authors use Seidel polynomials that appear to be a convenient choice because of the presence of field dependent astigmatism. Please explain this in the text.

We thank the reviewer for this helpful comment and advice. For applications that look at specific field positions, people commonly use Zernike modes to decompose the pupil plane wavefront profiles into different modes. But for the business of lens design, which needs to evaluate how the entire lens system behaves over the entire design FOV, the general aberration function contributed by Roland V. Shack, which describes aberration function using the field and aperture vectors, is the most used method. As the goal of this work is to achieve aberration correction over the entire FOV, we also followed this convention to describe the aberration of the system, which varies over the entire FOV. To convey the information to readers familiar with Zernike modes, we have added Supplementary Fig. 2 which lists the Zernike coefficients of each mode before and after GTAO correction for a field position 140 micron from the center of the FOV.

3. It is well known that there are different aberrations in GRIN lens. For example, in Fig. 2 of Ref. 43, the contribution of spherical aberration is the same as astigmatism. Furthermore, coma is also one of the main aberration sources. The authors need to explain whether their design compensates for other aberrations, or works only for astigmatism.

We thank the reviewer for the helpful comments and questions. We fully agree that there are various types of aberration. But for GRIN lens-based deep brain imaging, the dominant one (the image-killing aberration) is fourth-order astigmatism. As shown in Fig. 1a, there is still residual aberration after the GTAO correction, but it does not affect the goal of achieving large FOV *in vivo* volumetric recording, as demonstrated by the *in vivo* imaging comparison in Fig. 3 and Fig. 5. To make this clear to the general readers, we have added Supplementary Fig. 2 to quantify the contribution of various types of aberration, listed the Zernike coefficient for each term before and after GTAO correction, and compared the resulting Strehl ratio due to astigmatism, spherical aberration, and coma. Overall, despite the residual aberrations, GTAO has achieved the goal of large FOV volumetric recording through the slim 0.5 mm GRIN lens with whole-mouse brain penetration capabilities.

4. Please evaluate how much aberration is contributed by the inherent birefringence of the GRIN lens (He et al., Nat. Comm. 10, 4264 (2019)), and comment on whether this dual GRIN lens design is able to correct the intrinsic polarization/phase aberrations.

We thank the reviewer for the valuable question. As discussed earlier, even for the visible wavelength, the birefringence GRIN lens is a very weak effect. One will need GRIN lenses near ~100 mm lengths to experience such effects for the visible laser beam. In comparison, the widely employed GRIN lenses in neuroscience applications are only 6-7 mm long and the wavelength is in the near infrared. Thus, birefringence has a negligible impact on such applications. To make this point clear to the readers, we have added a supplementary Fig. 1, in which we showed the experimentally measured aberration profile with a linearly polarized laser input. As expected, the transmitted aberration profile is rotationally symmetric regardless of the polarization direction.

Another perspective is from the images obtained with GTAO correction. The linearly polarized laser was used in all experiments. If the birefringence is dominant, we would expect to see different PSF in different

regions of the FOV (e.g. 3 o'clock vs 6 o'clock), which however was not evident in both the beads and animal imaging results (e.g. see Figs. 1e, 2c, 3b, 5c).

5. The volumetric imaging results are impressive. However, when performing volumetric imaging under a GRIN lens, the magnification of each layer is often different. A common situation is that as the focus moves up, the image undergoes contraction around the FOV center, i.e. image distortion in the axial direction. The authors shall specify whether the results have been post-processed, or whether the dual GRIN lens design is able to tackle the axial distortion.

We thank the reviewer for the insightful comment. For GRIN lenses of a non-integer number of 0.5 pitch (e.g. 1.4 pitch, 1.9 pitch), the magnification through the GRIN lens may vary as a function of working distance. However, for GRIN lenses of an integer number of 0.5 pitch, such effects are minimized. In this work, we mostly used 1.5-pitch GRIN lenses, and the field of view variation is very small. As a result, we did not need to digitally adjust the images before assembling them into the final image volume. To provide supporting data, we performed new experiments using bead samples, in which we adjusted the WD and the sample position such that we imaged the same beads over a 350×350 micron² FOV at each WD using the same galvo control signals. We added Supplementary Fig. 6 to show the images. From the beads' positions near the boundaries, we can see that their spatial locations were consistent from 100 to 400-micron WD.

For applications that use GRIN lenses of a non-integer number of 0.5 pitches (e.g. 1.4, 1.9 pitches), the analog galvo control signal for each plane needs to be scaled to yield a consistent field FOV if the user desires. Without changing the Galvo scanning range, the FOV will vary over FOV. We also performed new experiments using 1.4 pitch GRIN lenses to show such cases and added the results to the new supplementary Fig. 7.

6. In Fig. 1a, the authors mentioned “astigmatism ...through the first GRIN lens reached 4.6 waves from the peak to the valley.” It is more informative if the authors provide Zernike or Seidel order analysis, in order to confirm that the 4.6 waves aberration is mainly caused by astigmatism.

We thank the reviewer for the insightful questions and advice. We have added a new Supplementary Fig. 2 to decompose the aberrations by type and show the impact of these other types of aberrations. Overall, the dominant one (the image-killing one) is the fourth-order astigmatism.

7. The specifications of most optical components are missing, e.g. objectives, SLM, galvo, etc. Note that in p. 6, the sentence “positioned a water immersion observation objective lens (Olympus 60x NA 1.2) under the GRIN lens” causes the confusion of whether the NAs of the objectives and the GRIN lenses match each other.

We thank the reviewer for the valuable comments and advice. We have revised the manuscript to provide the product information. The NA of the laser focus through GRIN lens is ~ 0.4 . Thus, the observation lens just needs to have an NA above 0.4 and support water immersion (to mimic the index in tissue). We used the NA 1.2 water immersion lens in our lab for its availability but not for necessity.

8. In Fig. 3b, the neurons outside the 280- μm diameter are clearly visible, but in Fig. 3h, the outside region disappears. What happened to the region outside the 280- μm diameter?

We thank the reviewer for the insightful comment. For these studies, we used EOM to turn off the laser beam outside the diameter. But actually, for soma resolution recording, the FOV can go beyond 280 microns, as demonstrated in Fig. 3. We have performed new experiments and revised imaging data in Fig. 3 (*in vivo* structural imaging) and Fig. 5 (*in vivo* calcium imaging) to show the even larger FOV (~ 330 microns).

9. In p. 9, “we performed a digital pixel shifting in the axial direction (Fig. 4c, d).” Please explain the detail of the “digital pixel shifting” process, which might link to my comment 3.

We thank the reviewer for the insightful question. With GTA0, there is still the field curvature of the GRIN lens. In other words, the image plane recorded contains curvature. For applications that require precise spatial registration between *in vivo* (through GRIN lens) and *ex vivo* (through the objective lens without field curvature) recording, we need to digitally reshape the recorded volume so that they can match the *ex vivo* data spatially. So the pixel shift is only along the z direction. There is no shift in the transverse direction as 1.5 (integer number of 0.5) pitch GRIN lenses were used in this measurement.

10. Given that a series of optical sections of a pyramidal cell is illustrated in Fig. 5g, it would be more persuasive if 3D reconstructed dendritic structures of neurons were presented.

We thank the reviewer for the valuable comments and advice. GTA0 is designed for rapid volumetric two-photon calcium imaging through GRIN lenses. For the majority of such studies, soma resolution recording is sufficient. Nevertheless, GTA0 can also record dendritic structures. We have performed new experiments to image tissue with more dendritic structures. We have added a new Supplementary Fig. 7 to show the imaging results. With GTA0, the dendritic structures were visible over the entire FOV. Without GTA0, they were only visible in the central FOV and became blurry in the outer regions.

11. P.3, line 3, prims  prism

We thank the reviewer for pointing out this typo, which has been corrected. Thanks!

Overall, we thank both reviewers for their insightful comments and advice, which helped us to improve the quality of our work and the clarity of the presentation. Thanks!

Reviewer #1 (Remarks to the Author):

I appreciate the authors for satisfactorily addressing all the concerns. On a minor note, I observed that the links for the Code and Data availability sections in the manuscript appear to be missing. Aside from this, I have no other concerns.

Reviewer #2 (Remarks to the Author):

The authors have satisfactorily addressed the comments from both reviewers, and I support the publication of this work in Nature Communications.